# Combined Diagnostic Accuracy of Diffusion and Perfusion MR Imaging to Differentiate Radiation-Induced Necrosis from Recurrence in Glioblastoma

**DOI:** 10.3390/diagnostics12030718

**Published:** 2022-03-15

**Authors:** Ankush Jajodia, Varun Goel, Jitin Goyal, Nivedita Patnaik, Jeevitesh Khoda, Sunil Pasricha, Munish Gairola

**Affiliations:** 1Department of Radiology, McMaster University, Hamilton Health Sciences, Hamilton, ON L8V 5C2, Canada; 2Department of Medical Oncology, Rajiv Gandhi Cancer Institute and Research Centre, Delhi 110085, India; 3Department of Radiology, Rajiv Gandhi Cancer Institute and Research Centre, Delhi 110085, India; jitin27031992@gmail.com (J.G.); jeeviteshkhoda@gmail.com (J.K.); 4Department of Laboratory & Histopathology, Rajiv Gandhi Cancer Institute, Delhi 110085, India; patnaik.nivedita@rgcirc.org (N.P.); drsunilpasricha@yahoo.com (S.P.); 5Department of Radiation Oncology, Rajiv Gandhi Cancer Institute, Delhi 110085, India; gairolam@hotmail.com

**Keywords:** glioblastoma multiforme, diffusion-weighted imaging, perfusion weighted imaging, radiation-induced necrosis, clinical decision-making

## Abstract

We aimed to use quantitative values derived from perfusion and diffusion-weighted MR imaging (PWI and DWI) to differentiate radiation-induced necrosis (RIN) from tumor recurrence in Glioblastoma (GBM) and investigate the best parameters for improved diagnostic accuracy and clinical decision-making. Methods: A retrospective analysis of follow-up MRI with new enhancing observations was performed in histopathologically confirmed subjects of post-treated GBM, who underwent re-surgical exploration. Quantitative estimation of rCBV (relative cerebral blood volume) from PWI and three methods of apparent diffusion coefficient (ADC) estimation were performed, namely ADC R1 (whole cross-sectional area of tumor), ADC R2 (only solid enhancing lesion), and ADC R3 (central necrosis). ROC curve and logistic regression analysis was completed. A confusion matrix table created using Excel provided the best combination parameters to ameliorate false-positive and false-negative results. Results: Forty-four subjects with a mean age of 46 years (range, 19–70 years) underwent re-surgical exploration with RIN in 28 (67%) and recurrent tumor in 16 (33%) on histopathology. rCBV threshold of >3.4 had the best diagnostic accuracy (AUC = 0.93, 81% sensitivity and 89% specificity). A multiple logistic regression model showed significant contributions from rCBV (*p* < 0.001) and ADC R3 (*p* = 0.001). After analysis of confusion matrix ADC R3 > 2032 × 10^−6^ mm^2^ achieved 100% specificity with gain in sensitivity (94% vs. 56%). Conclusions: A combination of parameters had better diagnostic performance, and a stepwise combination of rCBV and ADC R3 obviated unnecessary biopsies in 10% (3/28), leading to improved clinical decision-making.

## 1. Introduction

Despite multimodality treatment, a paltry 5-year survival rate of a mere 10% is seen in high-grade gliomas (HGG) [1] with a lack of standard definition of long-term survival in Glioblastoma multiforme (GBM) [2]. Initial treatment with surgical exploration involves removing the enhancing component of the tumor and further adjuvant treatment by radiation with or without a combination of chemotherapy [3].

Perfusion and diffusion-weighted MR imaging have been advocated for in the follow-up of Glioblastoma when differentiation of tumor recurrence from radiation-induced necrosis (RIN) is pivotal. As per NCCN recommendation, a follow-up MRI is performed every two months and then every three months if patients remained off treatment for more than a year [4]. During this crucial follow-up period, radiation-induced necrosis may occur; due to radiation therapy with a reported incidence of 3–24% [5,6]. Adjuvant radiation treatment causes increased capillary leakiness and alteration of the blood–brain barrier, amounting to augmented contrast enhancement on imaging. Histopathological representation confirms endothelial injury and fibrinoid necrosis [7] in this subgroup, instead of neoangiogenesis amounting to increased cellularity and vascular proliferation in the recurrent tumor. Perfusion imaging (PWI) captures this angiogenesis and vascular leakiness [8]. The non-enhancing necrotic component of the tumor contains liquefactive necrosis, whereas radiation-induced changes contain coagulative necrosis [9]. Diffusion restriction qualitatively estimates tissue microstructural environment through the visualization of the Brownian motion of water. Apparent diffusion coefficient (ADC) values are quantitative representatives of diffusion-weighted MR imaging (DWI).

There is a paucity of studies investigating the combined quantitative analysis of PWI and DWI to differentiate between recurrent tumors and RIN. Interestingly, the introduction of central diffusion [10] with its recent validation [9] is a feasible approach that can be put to use in regular clinical practice. We tried to discover the best combination of quantitative parameters obtained from DWI and PWI for improved diagnostic accuracy in the differentiation of recurrent tumors and RIN.

## 2. Materials and Methods

### 2.1. Subjects

After approval from the Institutional Review Board, a retrospective analysis of the PWI and DWI MR Imaging was performed at our institute, which included histopathologically confirmed GBM subjects who consented to surgery and adjuvant radiation treatment given either alone or in combination with chemotherapy between April 2016 and December 2019. On follow-up, these subjects presented with a suspicious lesion in contrast-enhanced MRI and underwent re-surgical intervention. We included forty-four consecutive patients after exclusion of subjects due to suboptimal MR imaging, including susceptibility artifacts (*n* = 6), without areas of visible necrosis (*n* = 4), and those who lacked Dynamic Susceptibility Contrast (DSC) perfusion imaging (*n* = 9). As per institutional protocol, MR imaging was performed within 72 h before surgical intervention.

### 2.2. MRI Technique

All examinations were performed on a single scanner—Siemens Avanto 1.5 Tesla MR Unit using a head coil. Conventional MRI imaging protocol includes axial T1 and post-contrast T1 (TR = 550 ms, TE = 8.4 ms, flip angle = 90°), axial T2 (TR = 5000 ms, TE = 90 ms, flip angle =150°, thickness = 5 mm), axial FLAIR (TR = 9000 ms, TE = 88 ms, TI = 2500 ms, thickness = 5 mm), and a post-contrast 3D fast low-angle shot (FLASH) brain examination (TR = 28.4 ms, TE = 4.7 ms, flip angle = 25°). The contrast was used in the form of IV injection of 0.1mmol/kg of gadobutrol (Gadovist 1.0; Bayer Schering Pharma, Berlin, Germany) and gadobenate dimeglumine (MultiHance Bracco, Milan, Italy). Volumetric T1-weighted inversion recovery spoiled gradient-echo sequences were used for contrast enhancement and central necrosis. DWI with 3-direction axial EPI sequences (TR/TE = 13,800/80.2 ms, section thickness = 2.5 mm, FOV = 25 × 22.5 cm, b = 1000 s/mm^2^, number of excitations = 4). Diffusion images were used to make ADC maps.

Perfusion weighted imaging includes Dynamic Susceptibility imaging (DSC), performed by echo-planar technique with the following parameters: TR = 1500 ms, TE = 30 ms, flip angle = 90°, slice thickness = 5 mm, matrix = 256 × 256, section thickness = 5 mm. Multisection image data were acquired every second for a total of 50 s with the contrast injection [5-mL/s bolus injection] beginning at 8 s, resulting in a total time just below 2 min. Post-processing, DSC images were transferred to the perfusion application in Siemens workstation. Using AIF (arterial input fraction) in the middle cerebral artery by choosing four or more of the best times, the graphs with significant T2* signal drop time ranges were manually adjusted with three time-points: First, at the start of the baseline; Second, at the beginning of the drop (contrast entry); and Third, at the peak of the recovery.

Contrast material leakage correction was executed on DSC images by using an earlier adopted method [11,12], which assumes an intact blood–brain barrier after passage and recirculation of gadolinium-based contrast material. The model assumes that the T2* signal intensity returns close to the baseline value. In areas of the compromised blood–brain barrier, the leakage of the contrast material marks as local T1 shortening, and subsequently, the signal intensity in regions of contrast leakage overshoot this value. Using the said model, enhancing voxels were selected, leakage coefficient was calculated, and the coefficient was subsequently used to correct the CBV for contrast material leakage. CBV (cerebral blood volume), CBF (cerebral blood flow), MTT (mean transit time), and TTP (time to peak) colored maps were generated. This corrected mean CBV was calculated for all enhancing voxels on the perfusion images. Relative cerebral blood volume (rCBV) maps were generated by comparison with contralateral white matter.

### 2.3. Imaging Analysis

Trained neuro-radiologists, each with more than seven years of experience, autonomously evaluated images. ADC maps were generated on Siemens workstation. ADC values were calculated manually by placing a region of interest (ROI) over the lesion using Pixel-wise ADC maps at a high b value of 1000 s/mm^2^. The readers were sentient that subjects under analysis were post-treated GBM patients but were blinded to histopathology results. The first reader (ADC R1) plotted a freehand ROI on ADC maps over the lesions whole cross-sectional area, including the solid enhancing part and non-enhancing necrotic part. The second reader (ADC R2) drew a standard ROI (using an area of at least 15 mm^2^, up to a maximum of 30 mm^2^) in the darkest part of the tumor region (DPTU) on ADC maps, corresponding to the zenith of diffusion restriction and the solid enhancing portion of the lesion, carefully excluding areas of hemorrhage and necrosis. A third reader (ADC R3) plotted freehand ROI manually, including each lesion’s necrotic component, carefully excluding the enhancing solid portion. Definition of necrosis was a non-enhancing region with fluid signal intensity surrounded by contrast enhancement. The mean ADC values were recorded for all observations. After an independent assessment of the lesions by all readers ADC (R1, R2, and R3), the final consensus to mark and draw the ROI was decided in congruence with two senior radiologists with more than 25 years of experience. In all cases, mean ADC was used.

Quantitative rCBV values were calculated manually on post-processing DSC PWI imaging sequences, congruence, the senior radiologists placed ROI on rCBV maps over areas of lesions representing high blood volume. Placement of ROI was performed post-analysis of the most suspicious area of enhancement, excluding areas of hemorrhage on post-contrast T1 weighted sequences using the auto-synchronization function in Siemens workstation.

### 2.4. Histopathology Interpretation

The final diagnosis as a recurrent tumor or RIN was assigned based on the histopathology results. As per institutional protocol, the presence of any tumor cells in the surgical specimen qualified the sample as a recurrent tumor regardless of the mixture of fractions in which tumoral and inflammatory cells were present. The blocks were reviewed independently by two senior pathologists with more than 15 years of experience, who were blinded to the radiological findings. Each case was assigned to another senior pathologist with over 30 years of experience, and the final opinion was reached in consensus.

### 2.5. Statistical Analysis

Statistical analysis was completed using MedCalc 15 (MedCalc software bvba, Ostend, Belgium). The diagnostic accuracy of individual parameters was evaluated by area under the receiver operating characteristic (ROC) curve (AUC). Optimal thresholds were determined for each ROC curve to maximize sensitivity and specificity using the Youden statistic. We also compared the statistical significance between differences in area under the curve (AUC) of individual parameters. Further specificity and sensitivity were tabulated in an Excel sheet, and confusion matrix tables were generated to find an optimal combination of parameters yielding the highest possible diagnostic accuracy. Sensitivity, specificity, and likelihood ratio values were obtained at different cut-off values. A forward logistic regression model was used to recognize the contribution of each parameter to distinguish recurrent tumors from RIN and 5-fold cross validation was carried out to calculate the mean ROC curve. *p* values < 0.05 were considered statistically significant.

## 3. Results

### 3.1. Baseline Characteristics

Sixty-three patients diagnosed with GBM had a new enhancing observation on follow-up MR imaging. After exclusion criteria, the final analysis of MR imaging was carried out in a cohort of forty-four eligible patients (14 women and 30 men) with RIN in 28 (67%) and recurrent tumor in 16 (33%) on final histopathology. The mean age at re-explorative surgery for new enhancing observation was 46 years (range, 19–70 years). The range and median of time duration from last day of radiation to MR Imaging with new enhancing observation were 4–190 weeks (Median = 20 weeks) for all lesions, 5–148 weeks (Median = 20.5 weeks) in recurrent tumor and 4–190 weeks (Median = 17 weeks) in RIN.

### 3.2. Quantitative MR Imaging Parameters with ROC Curve Analysis

We evaluated four quantitative parameters, namely, rCBV, ADC-R1 (Whole lesion including necrosis), ADC-R2 (Only enhancing solid portion), and ADC-R3 (Only central necrosis portion). The details of the range and median of all these four parameters have been provided in a tabulated manner in Table 1. There was a statistically significant difference in rCBV, ADC R1, and ADC R3 between the subgroups of recurrent tumor and RIN. Figure 1 shows an example of MR including PWI and DWI of a patient with RIN (A–E) and recurrent tumor (F–J).

The ROC curve analysis was completed for all these parameters to find the optimal cut-off criterion as determined by the largest sum of sensitivity and specificity (Ref Table 2). The individual parameter with the best diagnostic performance for differentiation of RIN from the recurrent tumor was rCBV at a threshold of >3.4, which had an AUC of 0.93 with 81% sensitivity and 89% specificity. The comparison of ROC curves to determine the best single parameter with the highest diagnostic accuracy is shown in Table 3 and Figure 2 and Figure 3. Logistic regression showed significant contributions from rCBV (*p* < 0.001; OR = 2.8) and ADC R3 (*p* = 0.001).

### 3.3. Combined Approach by Using Both PWI and ADC Parameters

Using optimum criteria of rCBV > 3.4 to predict recurrent tumor, we constructed a confusion matrix in Excel and ended up with three false negatives (FN) and three false positives (FP) cases. Utilizing the optimal criterion of ADC R3, we ended up with only one FN case. By choosing an optimal value of 2032 × 10^−6^ mm^2^/s in ADC R3, we achieved 100% specificity with increment in sensitivity (94% vs. 56%). Using the combined approach instead of stand-alone rCBV as a single best judge, we will obviate unnecessary biopsies in 10% (3/28) cases.

## 4. Discussion

We found the highest diagnostic yield by rCBV compared to any other parameter, and the addition of quantitative ADC parameters of solid enhancing portion to rCBV did not significantly improve the yield. In routine clinical practice, performing DWI is more feasible than PWI, as the former has rapid acquisition with translational benefits and does not need contrast administration. However, the addition of rCBV and ADC of central necrosis significantly improves performance and diagnostic accuracy.

Re-explorative surgery is invasive and adds to morbidity. If the combined approach correctly identifies the population where a biopsy can be averted, such an approach would be of practical significance in clinical decision-making. We improved the overall diagnostic performance with a combined cut-off >3.4 CBV that provided the best specificity (90%) and the addition of ADC-R3 central necrosis parameter of a threshold greater than 2032 × 10^−6^ mm^2^/s, increasing the specificity to 100%. This combined approach is the strength of our study and helps us with the incorrect classification of all 28 patients with RIN (compared with 25 of 28 patients when only rCBV was used). Analysis of ROC for rCBV with an optimal criterion cut-off >3.4 shows 90% specificity and 81% sensitivity, comparable with the previous literature [13,14]. Using the previous historical cut-off >1.75 [15,16], we would have two false negatives (FN) and seven false positives (FP) cases. Misclassification with rCBV has also been reported in past study [17] with 80% sensitivity, similar to ours, reinforcing the concept of co-existence of tumor cells infiltration in a background of coagulative necrosis and inflammatory infiltrates due to RIN.

Analysis of ROC for ADC-R3 (AUC 0.84; 95% CI, 0.7–0.93; *p* < 0.001), a cut-off ADC value of <2130 × 10^−6^ mm^2^/s yields 78% specificity and 68% sensitivity for the identification of RIN. This value is much higher than the first study documenting the importance of this central necrosis parameter [10]. A few reasons that could have led to this wide variation were the inclusion of a small cohort in the previous study (sample size almost a third of ours) and the classification of subjects with 5–20% tumor cell population into the RIN sub-group. A significantly higher diagnostic accuracy of the centrally reduced diffusion sign than ADC assessment of only the solid component was validated [9]. Analysis of ROC for ADC-R1 (area under the curve 0.82), a cut-off ADC value of <1416 × 10^−6^ mm^2^/s yields 78% specificity and 87.5% sensitivity for identifying identified recurrent tumors. This cut-off is similar to the previous studies [8] suggesting a cut-off ADC value of <1490 × 10^−6^ mm^2^/s (70% specificity and 73% sensitivity) and another study [17] predicting a cut-off ADC value of <1201 × 10^−6^ mm^2^/s (64% specificity and 78% sensitivity). Our study documented the highest sensitivity of this parameter (ADC R1) to date with the best combination of specificity but adding this parameter to rCBV did not improve the results, similar to the contribution by central restriction (ADC R3).

Selection of the most representative region of observation (DPTU) (ADC R2) did not statistically distinguish recurrent tumors from RIN, nor did combining this with rCBV improve diagnostic accuracy, as found in the previous study [5]. This is also congruent with the previous meta-analysis, which showed moderate diagnostic performance of DWI and recommended against the use of DWI alone in differentiating RIN and tumor recurrence [6]. Lower ADC values, which are usually expected in the tumor due to increased packed tumoral cells, may also confoundingly be less in RIN settings, likely due to infiltration of inflammatory factors and abundant polymorphonuclear lymphocytes [18].

This is not the first time that rCBV outperformed ADC in discriminating RIN [3,8,17,19,20]. Most of these studies had the limitations of a small cohort, mixed variety of low- and high-grade gliomas, inconsistent treatment with radiation, and lack of re-explorative surgery, thus heavily relying on tissue sampling, which had the potential of being under-sampled or non-representative. This problem may further be accentuated while dealing with a heterogenous entity such as GBM. Recent lessons from the literature have identified various semantic and explorative features from diffusion-weighted imaging (DWI) and perfusion weighted imaging (PWI) which have been helpful in the differentiation of recurrent tumors and RIN [8,10,20,21]. Few studies relied on radiomics and a deep learning approach rather than semantics for enhanced diagnostic performance in differentiation between actual tumor and RIN. Though the robustness of these approaches has been validated in a multi-center setting [22,23], practical reading room reporting cannot incorporate such methods.

Our study has potential limitations. First, the nature of the investigation was retrospective, with a third of our cohort diagnosed with recurrent tumor, possibly attenuating the analytical power of our study. This may partially be explained due to the poor compliance of subjects with recurrent tumors to adhere to the treatment protocol, leading to loss of follow-up. Second, like previous studies, we did not stratify our subgroups based on histopathology by a fraction of recurrent tumor versus the fraction of RIN due to lack of standard cut-off for defining RIN changes rather than the recurrent tumor. Lastly, our study design is case-controlled which may have limitations in representing the disease spectrum in a real clinical setting. The index tests’ cut-off value, including CBV, is likely to be domain-specific, and further external validation studies may only reflect its utility in the actual clinical situation. Previous research [24] documents 14–15% patients with conventional RT have cerebral radiation necrosis. New advanced radiation delivery techniques have ameliorated the incidence of necrosis. The documented brachytherapy rate of cerebral radiation necrosis is 25% to 50%. The gold standard for diagnosing cerebral radiation necrosis is necrotic lesion biopsy, which is costly, invasive, and subject to sampling inaccuracy. As such, exploiting newer MR imaging advances and using them in combination can yield results with fruitful clinical translational benefits. 

## 5. Conclusions

PWI and DWI are pivotal in differentiating recurrent tumors from RIN in Glioblastoma. Our results validate the previously documented role of rCBV and DWI parameters and provide significant clinical decision benefits through a combined approach, notably obviating unnecessary biopsies. Further large-scale prospective studies utilizing this combination to validate our findings will lead to infusion of these imaging parameters into clinical practice with imperative clinical decision-making outcomes.

Further, there was strict compliance with Ethical Standards.

## Figures and Tables

**Figure 1 diagnostics-12-00718-f001:**
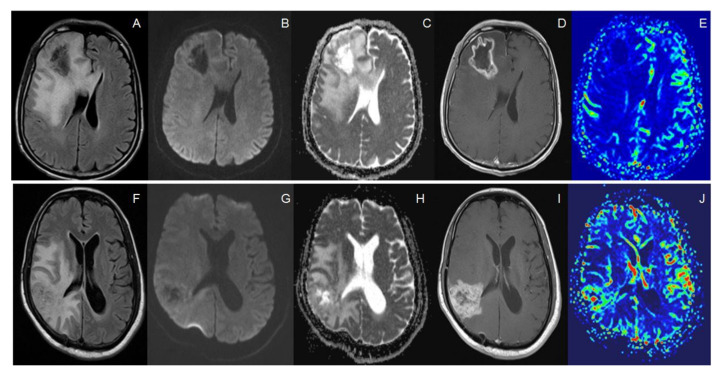
(**A**–**E**): Post op cavity with perilesional edema on FLAIR sequence (**A**), showing foci of diffusion restriction (**B**) with corresponding low ADC (**C**) and nodular peripheral margin enhancement (**D**). No abnormal areas of perfusion were seen (**E**). Histopathology showed radiation induced necrosis. (**F**–**J**): New irregular enhancement (**H**) with significant perilesional edema on FLAIR sequence (**F**), showing punctate foci of diffusion restriction (**G**) and abnormal areas of perfusion (**J**). Histopathology showed recurrent tumors.

**Figure 2 diagnostics-12-00718-f002:**
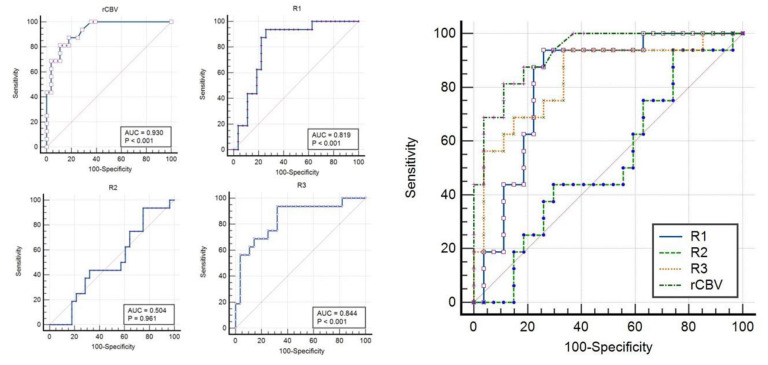
Receiver operating characteristic (ROC) curves with area under the curve (AUC) for both rCBV, ADC R1, R2, R3 (**left**) and comparison of ROC (**right**). Details of AUC with 95% CI and *p* values for comparison are provided in Table 3.

**Figure 3 diagnostics-12-00718-f003:**
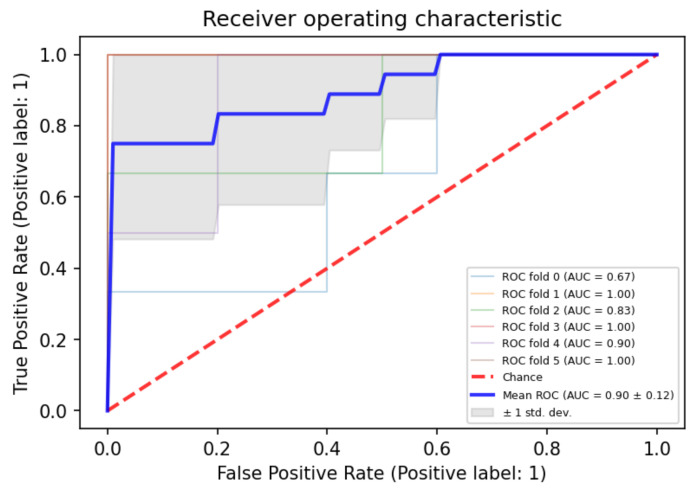
ROC curve of 5-fold cross validation tests for enhancers. (variance around mean curve is represented with gray shade representing confidence intervals).

**Table 1 diagnostics-12-00718-t001:** Details of MRI findings on MR Imaging with new enhancing observation.

MRI Parameters	Total n = 44 (%)	Tumor (*n* = 16) [36%]	RIN (*n* = 28) [64%]	*p*-Value
rCBV	1–14 (Median = 1.5)	1–14 (Median = 5.25)	1–5.6 (Median = 1)	*p* < 0.001
ADC-R1 (Whole tumor including necrosis) (expressed as values × 10^−6^ mm^2^/s)	900–2693 (Median = 1450)	991–1923 (Median = 1334)	900–2693 (Median =1639)	*p* < 0.001
ADC-R2 (Only enhancing solid portion) (expressed as values × 10^−6^ mm^2^/s)	536–2295 (Median = 1177)	825–2060 (Median = 1219)	536–2295 (Median =1155)	*p* = 0.709
ADC-R3 (Only central necrosis portion) (expressed as values × 10^−6^ mm^2^/s)	1379–3353 (Median = 2061)	1643–3353 (Median = 2603)	1379–2700 (Median =1843)	*p* < 0.001

**Table 2 diagnostics-12-00718-t002:** Details of ROC curve analysis with AUC of all parameters.

Variables	AUC	95% CI	Optimal Cut-off	Sensitivity	Specificity	*p*-Value
rCBV	0.930	0.811–0.985	>3.4	81.25	89.29	*p* < 0.001
ADC-R1 (Whole tumor including necrosis) (expressed as values × 10^−6^ mm^2^/s)	0.819	0.672–0.920	≤1416	87.50	77.78	*p* < 0.001
ADC-R2 (Only enhancing solid portion) (expressed as values × 10^−6^ mm^2^/s)	0.504	0.350–0.659	<536	0.00	100.00	*p* = 0.9608
ADC-R3 (Only central necrosis portion) (expressed as values × 10^−6^ mm^2^/s)	0.844	0.703–0.935	>2383	56.25	96.43	*p* < 0.001

**Table 3 diagnostics-12-00718-t003:** Comparison of ROC curves for both readings of all lesions with statistical *p* values.

Variables	AUC	95% CI	*p*-Value for ROC Comparison	*p*-Value for ROC Comparison	*p*-Value for ROC Comparison
rCBV	0.930	0.811–0.985			rCBV & ADC-R2 *p* < 0.001
ADC-R1 (Whole tumor including necrosis) (expressed as values × 10^−6^ mm^2^ /s)	0.819	0.672–0.920	ADC-R1 & ADC-R2 *p* < 0.001		
ADC-R2 (Only enhancing solid portion) (expressed as values × 10^−6^ mm^2^/s)	0.504	0.350–0.659		ADC-R2 & ADC-R3 *p* = 0.014	
ADC-R3 (Only central necrosis portion) (expressed as values × 10^−6^ mm^2^/s)	0.844	0.703–0.935			

## Data Availability

Data will be available upon request.

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
