# Peer review of "Combined Diagnostic Accuracy of Diffusion and Perfusion MR Imaging to Differentiate Radiation-Induced Necrosis from Recurrence in Glioblastoma"

_diagnostics, 2022, doi:10.3390/diagnostics12030718_

Round 1

Reviewer 1 Report

In thepresent paper the authors have the aim to use quantitative values derived from perfusion and diffusion-weighted MR imaging (PWI & DWI) to differentiate radiation-induced necrosis (RIN) from tumor recurrence in glioblastoma and investigate the best parameters for improved diagnostic accuracy and clinical decision making. Their conclusions consists in declaring that  the combination of parameters had better diagnostic performance and stepwise combination of rCBV and ADC R3 resolved unnecessary biopsies in 10 % (3/ 28) improving clinical decision-making.

The paper is well written and covers a topic of extreme importance in glioblastoma. The aim is centered perfectly on translational medicine and results gave per se an immediate use for the clinicians.

The combinations of several parameters is the way to go in such a heterogenous disease  as cancer.

I accept it with minor revisions

1) How many patients have a value of CBV over 3.4?

2) I would not dedicate an entire paragraph at the end of the paper to describe the limitations of this kind of diagnostics.  I would better  describe the advantages of using the combinations of the parameters to predict the diagnosis and authors should dedicate more space to explain how often they encounter these uncertain diagnostic cases and how crucial it is to find a way to avoid patient discomfort.

Reviewer 2 Report

To avoid overfitting and to assess the generalizability of the findings, cross validation is recommended. Having the limited data set, leave one out or 5-10 fold crossvalidation is recommended and cross validated ROC curves should be presented.

Author Response

5-fold cross validation ROC curve has been added as recommended

Line 178-184.

Round 2

Reviewer 2 Report

none